# Ultrafast nonthermal electron transfer at plasmonic interfaces

Yuying Gao [1,2,5] ✉, Jonathan Diederich [1,3,4,5], Yuxin Xie[2], Qianhong Zhu[2], Christian Höhn[1], Karsten Harbauer[1], Fengtao Fan [2], Can Li [2], Roel van de Krol [1,3] & Dennis Friedrich [1] ✉

Plasmon-induced charge generation and separation in metal/semiconductor heterostructures offer a promising platform for hot carrier-based energy conversion applications. A key challenge is understanding ultrafast hot carrier transfer at heterogeneous interfaces, as the details of plasmonic enhanced charge transfer dynamics and accompanying energy relaxation remain unclear. Here, by tracking charge transfer processes across spatial, temporal, and energy domains, we reveal ultrafast, nonthermal electron transfer directly from gold nanoparticles to gallium nitride (GaN) without energy losses from electron-electron scattering. This process facilitates efficient charge separation and produces a nonthermal distribution of transferred electron in GaN—contrasting with substantial energy dissipation typically observed during conventional interfacial charge transport. Furthermore, we demonstrate the pivotal role of light polarization in modulating charge generation and energy distribution, which enables dynamic control of electron relaxation and enhances the possibility of nonthermal electrons surmounting the Schottky barrier for successful injection. These insights pave the way for advancing hot-carrier management and achieving coherent control of non-equilibrium charge behavior across multiple dimensions for solar energy conversion and optoelectronic applications.

Plasmonic nanotechnology provides a promising possibility for controlling light-matter interaction in both the spatial and temporal dimensions[1,2], which has given rise to extensive scientific applications, ranging from optoelectronics[3] to photovoltaic devices[4] and photocatalysis[5,6]. Hot carriers in metallic nanostructures are at the heart of these applications[7], created through non-radiative plasmon decay on ultrafast timescales. Upon excitation of a surface plasmon, initially nonthermal carriers are generated and then interact with each other through electron-electron scattering, forming a thermal Fermi-Dirac distribution on timescales of hundreds of femtoseconds[8,9] with the electron temperature exceeding the lattice temperature. Electron-

electron and electron-phonon interactions on these ultrafast timescales result in a substantial loss of energy in hot carriers, which greatly constrains the efficiency of solar energy conversion and optoelectronic devices[10]. It is therefore crucial to characterize and understand the ultrafast behavior of hot carriers to fully explore their potential in plasmonic applications. This becomes particularly important in the field of plasmonic photocatalysis, where ultrafast and efficient charge separation is necessary to drive sluggish chemical reactions[11].

As a promising platform for plasmonic photocatalytic materials[12], hybrid structures with optimized Schottky barriers at the interface have been engineered to efficiently separate electron-hole pairs,

[1]Institute for Solar Fuels, Helmholtz-Zentrum Berlin für Materialien und Energie GmbH, Berlin, Germany. [2]State Key Laboratory of Catalysis, Dalian National Laboratory for Clean Energy, Dalian Institute of Chemical Physics, Chinese Academy of Sciences, Dalian, China. [3]Institut für Chemie, Technische Universität Berlin, Berlin, Germany. [4]Department of Chemistry, Princeton University, Princeton, NJ, USA. [5]These authors contributed equally: Yuying Gao, Jonathan Diederich. ✉e-mail: yuying.gao@helmholtz-berlin.de; friedrich@helmholtz-berlin.de

thereby extending the lifetimes of excited carriers[13]. This strategy opens up exciting possibilities for driving complex photochemical reactions, such as $CO_2$ reduction[14,15] and water splitting[16,17]. A key factor in optimizing photocatalytic performance is a better understanding of the transfer processes of plasmonic hot carriers across the metal/semiconductor interface. However, studying interfacial charge transport processes is challenging due to the ultrafast and intricate dynamics of charge transfer[8,18] and hot carrier cooling in metallic nanostructures[19,20]. This is further complicated by spatially heterogeneous interfaces that present hybrid chemical properties and localized electric field distributions[21]. Moreover, most studies of plasmon-induced charge transfer mechanism have relied on photocurrent measurements at the device level, or detection of total transferred charge density from nanoparticles into semiconductors[22–24], failing to discriminate between ballistic nonthermal carriers and thermalized carriers that have undergone electron–electron scattering events before interfacial injection. Although theoretical studies and computational modeling have attempted to reveal the energetic distribution and thermalization processes of hot carriers in metal nanostructures by making strategic assumptions about the dominant relaxation pathways[19,25], accurately incorporating the effects of ultrafast hot carrier transport and cooling in the presence of complex metal/semiconductor interfaces into computational models remains a significant challenge. As a result, much less is known about the energetic distribution of hot carriers during the ultrafast transfer across the metal/semiconductor interface, a better understanding of which could help elucidate the critical role of nonthermal electrons in charge separation across the plasmonic interface, which strongly impacts device performance. To unlock the potential of hot carriers for driving chemical reactions, it is essential to gain a detailed comprehension of how the electronic and chemical properties of the plasmonic metal/semiconductor interface determine the ultrafast hot electron transfer and cooling processes.

Here, by using femtosecond time-resolved two-photon photoemission (TR-2PPE) and surface photovoltage microscopy (SPVM) with nanometer resolution, we probe plasmon-induced hot electron relaxation and charge transfer dynamics in gold nanoparticles deposited on GaN substrate (Au NP/GaN). We observe ultrafast direct nonthermal electron injection at the Au NP/GaN interface with strong interactions occurring within the laser pulse duration (<40 fs) upon plasmon excitation, as evidenced by a nonthermal distribution of transferred hot electrons in the GaN conduction band and an extension of their occupation lifetimes beyond predictions via the Fermi liquid model. Furthermore, we examined the influence of pump polarization on the ultrafast charge transfer process and elucidated its critical role in governing the energetic distribution of excited electrons–an outcome difficult to attain in monodisperse plasmonic nanostructures lacking coupled hotspots, demonstrating a universal approach for precise control of hot carrier dynamics in future plasmonic technologies.

## Results

### Plasmonic properties and energy band alignment in Au/GaN

Plasmonic metal/semiconductor heterostructures utilizing the surface plasmon resonance (SPR) effect offer an efficient pathway for spatial charge transfer at the plasmonic interface[9,26] and enable the possible control of selective charge injection due to the formation of a Schottky barrier[23,27]. In the present work, we deposited Au nanoparticles (NP) of 7 nm on an n-type GaN substrate (Au NP/GaN, Supplementary Fig. 1). Au NP are selected to serve as plasmonic antennas because of their excellent light-matter interaction strength[28] and ability to efficiently transfer charge to underlying semiconductor[11,29]. The size and morphology of metal nanoparticles are known to influence the efficiency of hot-carrier injection[30,31], the Schottky barrier height[32], and the injection process[30]. In this study, we selected ~7 nm Au nanoparticles because

this size ensures sufficiently strong plasmonic excitation while remaining within the electron escape depth (~11 nm)[33] for the subsequent 2PPE studies, thereby allowing us to directly probe the ultrafast interfacial charge transfer dynamics at the Au/GaN interface. The transparent dielectric semiconductor GaN is chosen as the substrate due to its wide bandgap (3.4 eV, Fig. 1a), allowing for visible light excitation of SPR in Au NP without inducing photogeneration of electron-hole pairs directly within the GaN. In addition, GaN provides accessible conduction band states for plasmonic electron injection from SPR excitations[27]. Given the importance of interface interactions for photogenerated electron transfer, we also prepared Au films on the GaN substrate without subsequent annealing treatments for comparison. The Au film has a thickness of 2 nm and a roughness of less than 0.5 nm (Supplementary Fig. 2). The thickness of the Au film is smaller than the mean free path of photoexcited electrons (~10 nm)[34,35], ensuring that most photoexcited electrons reach the interface for injection. Absorption spectroscopy exhibits a SPR peak around 520 nm (2.38 eV) for Au NP/GaN (Fig. 1a). The oscillatory features observed at longer wavelengths are related to Fabry-Perot interference[23].

The nanoscale current-voltage curve measured centrally on the surface of Au NP (inset in Fig. 1b) using conductive atomic force microscopy (CAFM) depicts a characteristic of Schottky-like junction at positive potentials (Fig. 1b), arising from the work function difference between the Au NP and GaN (Supplementary Figs. 3, 4). We determined the Schottky barrier height to be 1.13 eV by fitting nanoscale I-V curve using the thermionic emission model (see details in Methods). An upward band bending of 0.16 eV is observed following Au deposition (Supplementary Figs. 5, 6). Based on these measurements, the energy band alignment of the Au NP/GaN system is shown in Fig. 1c. The GaN conduction band minimum (CBM) is located at −3.29 eV, the valence band maximum (VBM) at −6.69 eV, and the Fermi level at −4.42 eV relative to the vacuum energy level ($E_{vac}$). This Au/GaN band alignment filters by photon energy, allowing only hot electrons at sufficient energy to reach the GaN CBM to be injected across the interface.

### Mapping of the photo-induced charge distribution

We employ surface photovoltage microscopy (SPVM)[36,37] at the nanometer scale under steady-state illumination conditions to enable the direct visualization of photogenerated charge transfer. AFM image shows that the Au NP are dispersed individually on the GaN substrate (Fig. 1d), in contrast to the smoother Au film/GaN control sample (Fig. 1e). By subtracting the surface potential image acquired under dark conditions from that obtained under 520 nm illumination for SPR excitation (Supplementary Fig. 7), we observe a positive SPV signal at the spatial positions of the Au NPs (Fig. 1f). In contrast, no significant SPV signal (Fig. 1g) was detected on the Au film/GaN within the energy resolution of the instrument (5 mV). Figure 1h provides a quantitative comparison of the SPV distributions for both the Au NP and Au film samples under identical experimental conditions. It reveals effective photo-induced electron transfer to GaN in the Au NP/GaN sample, in contrast to the suppressed charge transfer observed in the Au film/GaN sample. Since both samples exhibit nearly identical absorbance, these differences in charge transfer behavior cannot be attributed to the SPR enhancement effect.

To understand the underlying reason for the distinct charge transfer behavior, we perform X-ray photoelectron spectroscopy (XPS) to investigate the chemical states and electronic interactions between Au and GaN. The high-resolution N 1s and Ga 2p XPS peaks for both Au NP/GaN and Au film/GaN shift to higher binding energies after Au deposition (Supplementary Fig. 8), suggesting electron transfer from GaN to Au. This results in the Au NP and Au film being negatively charged ($Au^{\delta-}$) relative to the GaN substrate. In the Au 4f spectra (Fig. 1i), the Au $4f_{5/2}$ and $4f_{7/2}$ peaks for Au NP/GaN are ~0.36 eV lower than those for Au film/GaN. Moreover, Au NP/GaN exhibits higher N 1s

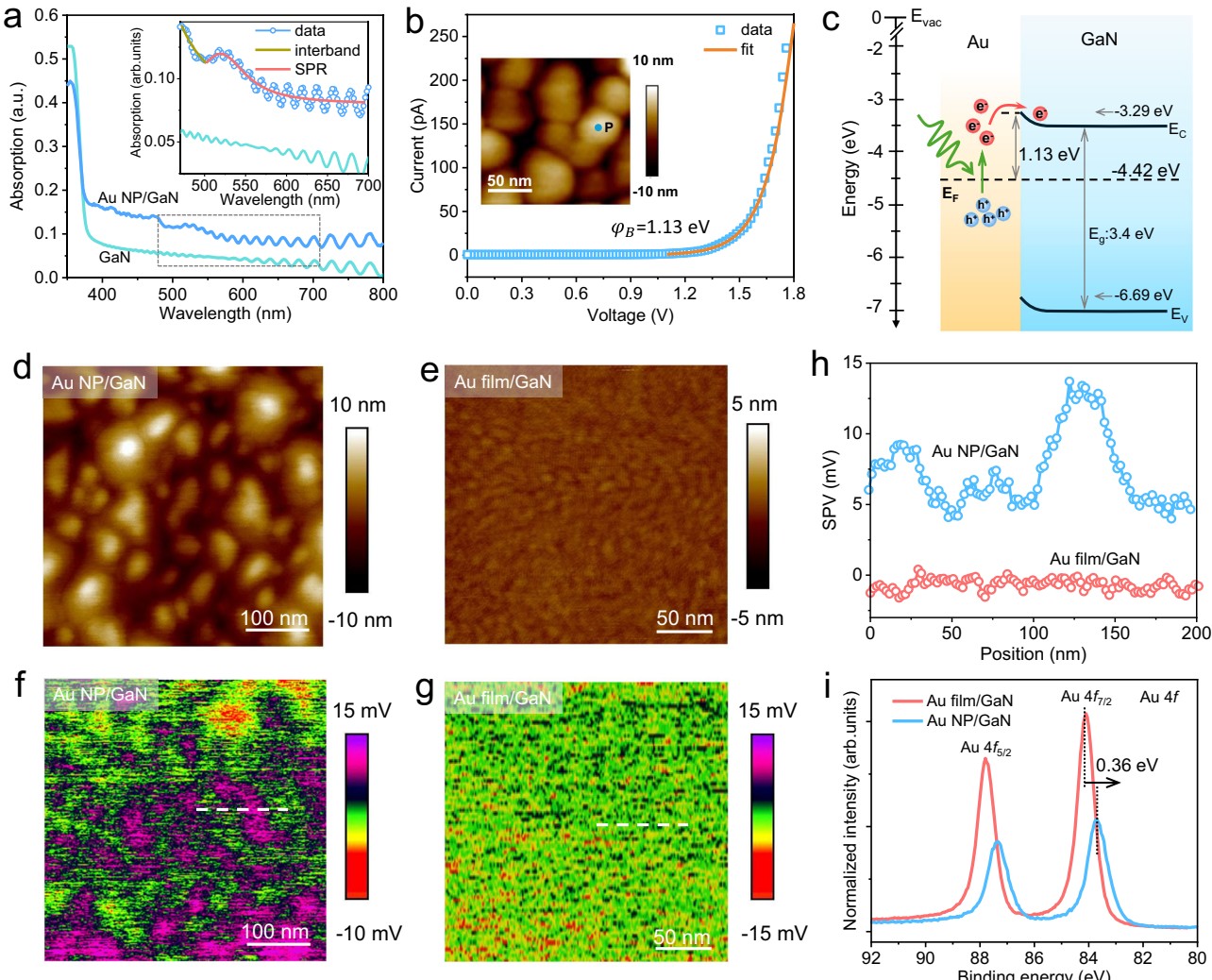

**Fig. 1 | Surface characteristics and imaging of interfacial charge transfer.**
**a** Absorption spectra of the bare GaN substrate and Au NP/GaN samples. The inset is a magnified view of the dashed line region, illustrating the contributions of the SPR peak at ~520 nm and interband transitions to the absorption. **b** Nanoscale solid-state current-voltage (I-V) curve of the Au NP/GaN sample, measured at the Au NP surface at point 'P' as indicated in the inset AFM image, using conductive AFM (CAFM). The solid line is a fit to the thermionic emission model. The Schottky barrier ($\varphi_B$) at the interface is estimated to be 1.13 eV. **c** Energy band alignment of Au NP/GaN heterostructure. The energies of the band gap ($E_g$), conduction band (CB), valence band (VB), and Fermi level ($E_F$) are determined by a combination of UPS, absorption spectroscopy, and CAFM. The green arrow represents the incident pump light, which excites electrons (red circle) within the Au NP. The red arrow indicates the transfer of photo-induced hot electrons across the interface to the CB of GaN. AFM topography of Au NP/GaN (**d**) and Au film/GaN (**e**). Surface photo-voltage (SPV) images of Au NP/GaN (**f**) and Au film/GaN (**g**) samples under excitation at 520 nm with an intensity of 5 mW/cm². **h** SPV line profile distributions of the two samples, taken along the dashed white lines in (**f** and **g**). **i** High-resolution XPS of Au 4*f* for the Au film/GaN and Au NP/GaN heterostructures.

and Ga 2*p* binding energies than Au film/GaN, indicating stronger electron transfer from GaN to Au in the Au NP/GaN system. This is indicative of strong interfacial interactions in Au NP/GaN compared to the unannealed Au film/GaN, due to intimate interface contact, which is a key factor that enables efficient charge transfer and injection at the interfaces[21,26,38]. These results imply that the stronger interaction of the Au NP with the GaN substrate could contribute to more effective charge separation of Au NP/GaN.

## Ultrafast direct nonthermal electron transfer
To obtain detailed information on the photo-induced interfacial electron transfer, we performed time-revolved two-photon photoemission (TR-2PPE) spectroscopy to probe the ultrafast behavior of photo-excited hot electrons[39,40]. We used a 2.38 eV (520 nm, VIS) pump pulse to resonantly excite SPR and a 4.33 eV (286 nm, UV) probe pulse to eject the resulting photoexcited electrons into the vacuum for detection. Figure 2a, b presents a pseudo-color plot of TR-2PPE spectra for

the two samples, which likely contain contributions from both photoelectron emission within the top Au NP and GaN surface states within the band gap. To clarify the origin of these signals, we performed TR-2PPE on pristine GaN under the same illumination conditions for comparison with the Au-modified surfaces. The bare GaN surface exhibits only a single time-independent emission peak at a kinetic energy of 0.44 eV (Supplementary Fig. 9). This peak is likely related to a surface state near the Fermi level, with emission occurring via one-photon photoemission (1PPE) using UV probe photons. This is substantiated by emission counts remaining independent of VIS pump fluence while scaling with UV probe fluence (Supplementary Figs. 10, 11). Notably, the 0.44 eV signal disappears after Au deposition, due to the surface metal layer suppressing photoemission from these states or passivating them[41]. This confirms that the TR-2PPE signal in Fig. 2a, b originates from photoexcitation of the deposited Au rather than the GaN substrate. Furthermore, the work function of Au NP/GaN (4.42 eV) is smaller than that of Au film/GaN (4.54 eV) samples, leading to a

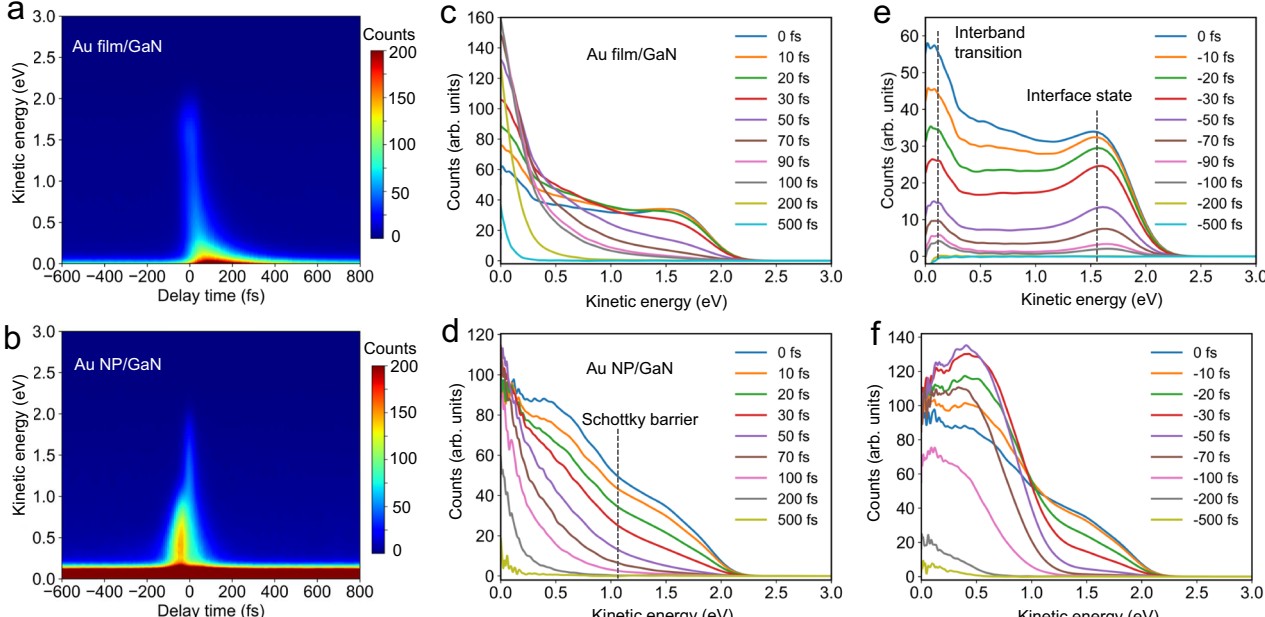

**Fig. 2 | Time-resolved 2PPE spectra. a** Representative pseudo-color plot of TR-2PPE spectra of Au film/GaN excited at 2.34 eV and probed at 4.33 eV. **b** Representative pseudo-color plot of Au NP/GaN excited at 2.34 eV and probed at 4.49 eV. The time-independent signal at a kinetic energy of ~0.1 eV corresponds to a one-UV-photon-induced 1PPE signal originating from the Fermi level. **c, d** 2PPE spectra of Au film/GaN (**c**) and Au NP/GaN (**d**) samples, extracted from **a** and **b** over a range of time delays from 0 fs to 500 fs after subtracting the time-independent

background signal. The dashed grey line in **d** indicates the energetic position of the Schottky barrier at the Au/GaN interface, which inhibits the injection of low-energy electrons. 2PPE spectra of Au film/GaN (**e**) and Au NP/GaN (**f**) samples, extracted over a range of time delays from 0 fs to −500 fs after subtracting the time-independent background signal. The dashed grey lines in (**e**) denote interband transitions and interface states, respectively.

strong time-independent emission around 0.1 eV for Au NP/GaN due to photoemission from near the Fermi level via both UV 1PPE and VIS 2PPE. Power-dependent 2PPE spectra show a linear relationship between photoelectron counts and laser fluence (Supplementary Fig. 12a, b), suggesting that the observed time-dependent emission originates from initially unoccupied states populated through sequential absorption of one VIS and one UV photon. The TR-2PPE spectra of Au NP/GaN were highly reproducible over repeated pulse cycles (Supplementary Fig. 12c), with no detectable decrease in intensity or change in spectral features, confirming the robustness of the interface under prolonged excitation.

By subtracting the time-independent background (taken at 3000 fs delay) from the full TR-2PPE data (Supplementary Fig. 13), we extract time-dependent pump-probe spectra at both positive (Fig. 2c, d) and negative (Fig. 2e, f) delay times for the Au NP and Au film samples, respectively. Positive delays time denote the VIS pump preceding the UV probe, and negative delays vice versa. For Au film/GaN, photoexcitation produces a nearly uniform distribution of nonthermal electrons above $E_F$ at 0 fs delay (Fig. 2c), extending up to kinetic energies corresponding to the pump photon energy (2.38 eV). This results from the uniform density of states (DOS) in the *sp* band below $E_F$ of Au[1]. Immediately after photoexcitation, these nonthermal electrons undergo rapid thermalization through electron-electron scattering within tens of femtoseconds[33], leading to a decrease in the population of high-energy electrons, along with an increase in the population of low-energy electrons[42]. As the delay time increases, the TR-2PPE spectra exhibit an exponential decay distribution, which is characteristic of thermalized hot electron distributions[41] that can be described by Fermi-Dirac statistics with an elevated electron temperature (discussed below)[43]. The electron population further cools via the dissipation of energy to the lattice through electron-phonon scattering within approximately 500 fs. In contrast, the strong electronic interactions at the Au NP/GaN interface favor charge injection

into the GaN substrate. This is evident in the two distinct kinetic energy ranges observed in the 2PPE spectra, with different 2PPE intensities and separated by a transition at the Schottky barrier at 0 fs (Fig. 2d). These findings suggest the presence of two distinct photoelectron emission mechanisms in Au NP/GaN.

In principle, the excitation of SPR is expected to generate a higher ratio of high-energy to low-energy electrons compared to bulk plasmons excited in the Au film at 0 fs delay[44]. However, this alone cannot account for the observed difference between the two samples. Another notable spectral feature of the Au NP/GaN samples is the lack of an increase in the population of low-energy electrons over time, which would typically be expected as electrons relax in metallic nanostructures[45,46]. This behavior is rather different from the Au film/GaN samples, where no significant interfacial charge transfer occurs, and the relaxation of electrons within the Au film causes an increase in low-energy electrons population over tens of femtoseconds. Therefore, the high-energy electrons with kinetic energies of emission above 1.1 eV in the Au NP/GaN can be attributed to photoelectron emission from the GaN substrate, following electron injection from the Au NP. Conversely, the low-energy electrons with kinetic energies below 1.1 eV are directly emitted from the Au NP (Supplementary Fig. 14a).

If hot electrons are generated in the Au NP and subsequently transferred to GaN via the widely proposed conventional indirect electron transfer mechanism[18], we would observe an increase in the 2PPE signal intensity from the GaN substrate within 100 fs, as the timescale for indirect charge transfer at the plasmonic interface is known to exceed 100 fs[10]. However, such an increase is not observed in Fig. 2d. Moreover, conventional indirect charge transfer is unlikely to produce a nonthermal distribution in GaN, as the electron-electron scattering events involved in this process would thermalize the electron distribution[41]. Therefore, the preservation of a nonthermal electron distribution in GaN up to the pump energy at 0 fs (Fig. 2d) strongly suggests ultrafast nonthermal electron transfer occurring

within the pulse duration (~40 fs), before significant e-e scattering dissipates their energy. This ultrafast transfer occurs on a timescale beyond the resolution of our measurement, consistent with recent observations of charge injection in Au/TiO$_2$ on a timescale of ~11 fs determined by terahertz emission[1]. Thus, the nonthermal electron transfer demonstrated here is distinct from the conventional indirect mechanism, which generally involves scattering-induced thermalization and slower energy-relaxed transfer[18]. Plasmon-induced hot electrons are primarily generated at the Au NP surface, where the enhanced near-field distribution[47] facilitates subsequent electron injection at the Au NP/GaN interface. Based on the size of Au NP and the laser pulse width, the transport velocity of hot electrons is calculated to be at least $v \approx 1.5 \times 10^5$ m/s, of the same order of magnitude as the ballistic transport velocity of excited nonthermal electrons in Au ($8.9 \times 10^5$ m/s, Supplementary Note 1)[48], suggesting that plasmon-induced electron transfer occurs primarily within the ballistic regime. To further confirm this, we performed simulations of the electron energy distribution accounting for scattering events, which show that charge transport proceeds with negligible influence from electron scattering ($n \leq 1$, Supplementary Fig. 15a and Supplementary Note 2), verifying the ballistic transport behavior. Such charge transfer process is predominantly governed by non-equilibrium electrons and is less influenced by thermalized and Auger electrons, as evidenced by the linear increase in emission counts of ballistic nonthermal electrons with laser fluence (Supplementary Fig. 12 and Supplementary Note 3).

To examine the effect of SPR excitation on emission signals, we compared the 2PPE spectra for the two samples within the same measurement session to avoid any differences in laser fluence and focus conditions across both samples (Supplementary Fig. 16a). The number of excited electrons is expected to be proportional to $|\mathbf{k} \cdot \mathbf{E}(r,t)|^2$, where $\mathbf{k}$ is the electron wave vector and $\mathbf{E}(r,t)$ is the plasmonic electric field distribution. Consistent with this, we found that the emission intensity of Au NP/GaN is higher than that of Au film/GaN due to plasmonic near-field enhancement in Au NP/GaN. Notably, the emission signal from transferred electrons in GaN remains stronger than that from the Au film, underscoring the efficient charge separation process. Further analysis of the energy distribution spectra reveals a charge injection efficiency of ~31% for Au NP/GaN (Supplementary Note 4), facilitated by direct ballistic nonthermal electron transport. This mechanism contrasts with conventionally reported pathways[49,50], where hot-electron transfer efficiency (<10%) is diminished by competition with ultrafast relaxation processes. Additional support for this mechanism is provided by the strong dependence of 2PPE intensity from injected electron on pump photon energies (Supplementary Fig. 16b). The emission intensity at 0 fs is maximized when excitation occurs near the SPR wavelength, highlighting the importance of surface plasmons in extracting nonthermal electrons and driving the ballistic nature of the transport processes.

At negative delay times, two distinct emission peaks are observed in the Au film/GaN at 0.10 eV and 1.62 eV (Fig. 2e). The first peak can be attributed to transitions from the $d$ to $sp$ band in Au due to the relatively higher probability of direct interband transitions under UV photoexcitation compared to contributions from indirect intraband transitions[51]. The second peak (1.62 eV) originates from interface states distributed at 0.65 eV below the $E_F$ (Supplementary Fig. 17 and Supplementary Note 5). For the Au NP/GaN, the interface states become more pronounced at pump energies exceeding 2.48 eV and can be emitted by two VIS photons, producing a time-independent signal (Supplementary Figs. 18, 19). We fitted and calculated the energetic positions of these interface states located at 0.53 eV and 0.75 eV below the $E_F$ (Supplementary Note 5). Under UV exposure, hot electrons generated through both interband and intraband transitions possess sufficient energy to overcome the Schottky barrier for injection, which is supported by the increase in emission intensity around 0.40 eV within 50 fs due to the relaxation of high-energy transferred electrons

(Fig. 2f) in the GaN conduction band. These results suggest that Au NP/GaN supports ultrafast nonthermal electron transfer even under UV excitation (Supplementary Fig. 14b).

## Hot electron relaxation dynamics

To uncover the effect of ultrafast charge transfer on hot electron relaxation behavior, we compared the time-dependent photoemission signal at various kinetic energies for Au film/GaN (Fig. 3a) and Au NP/GaN (Fig. 3b). Following photoexcitation, the decay of hot electrons at all kinetic energies in both samples occurs on a timescale slower than the excitation pulse duration (top panels of Fig. 3a, b). For the Au film/GaN, the photoemission peak shifts towards positive delay times at lower kinetic energies as a result of the finite rate of high-energy electron relaxation. In contrast, the photoemission peak for Au NP/GaN samples shifts to negative delay times at low kinetic energies (<1.1 eV), implying a slower growth process stemming from the thermalization of high-energy electrons in GaN during negative delay times.

By fitting the experimental data with a Gaussian response function convolved with an exponential decay (Supplementary Note 6), electron lifetimes are extracted and plotted as a function of energy versus the Fermi level (E-E$_F$), as shown in Fig. 3c. Theoretical predictions for the energy dependence of relaxation time can be modelled as[52] $\tau^{-1} = \tau_0^{-1} + \alpha(E - E_F)^\beta$, where $\tau_0$ is a constant accounting for contributions from electron transport and electron-phonon scattering, the second term represents the contribution from electron-electron scattering according to Fermi-liquid theory (FLT), and $\alpha$ describes the probability and strength of electron-electron interactions. Fitting experimental data for Au film/GaN (dotted grey line in Fig. 3c) using this model yields $\tau_0 = 124 \pm 10$ fs and $\beta = 1.93 \pm 0.06$. The contribution of electron transport in the Au film to the emission is likely to be negligible due to the Au film thickness (2 nm) being significantly smaller than the photoelectron escape depth in Au (~11 nm)[33]. The observed $\tau_0$ therefore predominantly reflects the influence of electron-phonon scattering on hot electron decay pathways and is of the same order as previously reported values for Au films[33]. The fitting result for $\beta$ agrees well with the theoretical value of 2 based on FLT, indicating that electron-electron scattering dominates the hot electron relaxation process[29,53]. A similar fitting procedure was performed for Au NP/GaN for excited electron energies below the Schottky barrier height (1.13 eV), corresponding to photoemission from Au. The resulting fit yields $\tau_0 = 197 \pm 20$ fs and $\beta = 1.60 \pm 0.14$. Compared to Au film/GaN, the lower $\beta$ and higher $\tau_0$ values for Au NP/GaN can be ascribed to back-injection of electrons to attain charge compensation, occurring on a timescale of ~120 fs (Supplementary Note 7) following thermalization with the GaN lattice[54].

As a reference for the hot electron dynamics, we also prepared a control sample by inserting an insulating Al$_2$O$_3$ interlayer at the Au/GaN interface (see Methods for details). TR-2PPE measurements show that the presence of Al$_2$O$_3$ does not alter the initial energy distribution of hot electrons but accelerates their energy relaxation (Supplementary Figs. S20, 21). This effect arises from the suppression of interfacial charge transfer, which increases the hot-electron population within Au and thereby enhances electron–electron scattering[55], leading to faster cooling and shorter lifetimes.

Another key observation is that the lifetimes of injected electrons with kinetic energies above 1.13 eV in Au NP/GaN are longer than predicted values based on FLT for metallic systems. The anomalously extended lifetime is unlikely to result from Auger scattering associated with d-band hole relaxation, as this process is known to be weaker in Au[33] compared to Cu[56]. Indeed, we did not observe an Auger contribution to the electron lifetimes for Au film/GaN. Therefore, the prolonged lifetime can be attributed to electron relaxation in GaN[54] following ultrafast nonthermal electron transfer at the interface[26,57].

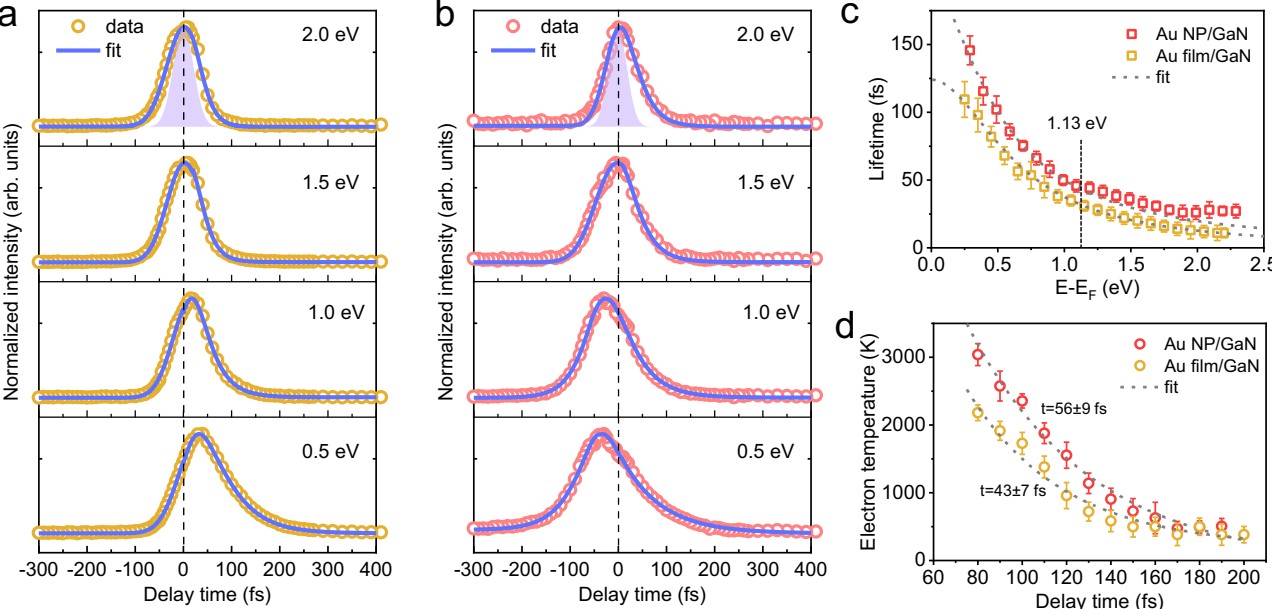

**Fig. 3 | Hot electron thermalization dynamics. a** Energy-resolved photoelectron intensity as a function of delay time for Au film/GaN samples excited at 2.34 eV; the yellow circles are experimental data points. The decays are fitted with the kinetic model as described in Supplementary Note 6. **b** Energy-resolved photoelectron intensity as a function of delay time for Au NP/GaN excited at 2.34 eV. The red circles are experimental data points, and the solid lines fit the equation in Supplementary Note 6. The purple shaded Gaussian areas in **a** and **b** represent the pump-probe cross-correlation obtained from an occupied state of Cu (111).

**c** Lifetimes of excited carriers as a function of electron energy relative to E-$E_F$ for both Au NP and -film samples. Lifetimes are extracted from the fitting parameters of the plots in **a** and **b**. The dashed grey lines are fits to a Fermi-liquid model, and the Schottky barrier is indicated as a dashed line. **d** Time-dependent electron temperature extracted from fits of the higher energy tails of 2PPE spectra to Fermi−Dirac distributions for both samples. The dashed grey lines are fit to an exponential decay to account for the time constants. Error bars in **c** and **d** represent the standard deviation of the respective fits.

To understand the relaxation dynamics of hot electrons in GaN following injection, we conducted TR-2PPE experiments across a range of pump fluences to examine the correlation between charge carrier lifetimes and charge densities. As shown in Supplementary Fig. 22, the dynamics of transferred electrons in GaN are only weakly affected by laser fluence, excluding the possibility of significant contributions from electron trapping and electron-hole scattering in GaN[58]. Moreover, intervalley scattering within the GaN substrate was not possible as the excess energy of transferred electrons was less than the intervalley threshold energy of 1.34 eV for GaN[59]. Raman spectroscopy reveals a higher longitudinal optical (LO) phonon energy (~91 meV, Supplementary Fig. S23) for GaN compared to materials, such as perovskite (25 meV)[60] and Si (59 meV)[61], resulting in faster electron–phonon scattering times in GaN (~18 fs, Supplementary Note 8). These findings indicate that hot electron cooling in GaN is primarily governed by LO phonon-electron interactions, consistent with previous theoretical predictions[62]. The observed hot electron lifetimes in GaN are on the order of tens of femtoseconds, comparable to those reported for other plasmonic heterostructures, such as Ag/TiO2[41] and Ag/graphite[63].

By fitting the high-energy tails of the 2PPE spectra (Fig. 2c,d) with a Fermi−Dirac distribution (Supplementary Note 9), we reconstructed the temporal evolution of electron temperatures for both Au NP and Au film samples, enabling a detailed investigation of the influence of SPR on hot electron cooling. The fitted data starting at around 80 fs reveal the emergence of a quasi-equilibrium distribution consistent with Fermi-Dirac statistics, indicating a thermalized carrier distribution (Fig. 3d). As demonstrated above, electron-phonon interactions drive the rapid decay of electron temperature from ~3000 K to ~400 K within ~200 fs in Au NP/GaN. The initially higher electron temperature in Au NP/GaN compared to Au film/GaN arises from the increased absorbed energy density[23] and the efficient ultrafast nonthermal

electron transfer in Au NP, both governed by surface plasmon effects. Consequently, hot electron cooling in Au NP occurs at a relatively slower rate than in Au films, in agreement with the predictions of the two-temperature model[54].

## Polarization dependence of ultrafast nonthermal electron transfer

Given the well-established polarization sensitivity of charge transport trajectories of nonthermal electrons and plasmonic field distribution[1], we further examine how nonthermal electron transfer depends on the laser polarization angle. An illustration of the pump and probe beams incident on a sample and photoemitting electrons is given in Fig. 4a, specifically for the Au NP on GaN case. Both pump and probe beams are initially *p*-polarized relative to the sample. The angle (θ) of the pump beam relative to the *p* polarization is controlled by rotating a polarizer for a fixed incident angle (45° from the surface normal). This experiment is performed at a pump photon energy of 2.43 eV (510 nm), combined with a probe photon energy of 4.34 eV (286 nm). The pump polarization is advanced in steps of 20°, with additional measurements at 90° and 270°. Figure 4b depicts emission at kinetic energies around 2.0 eV at 0 fs that is related to nonthermal electron transfer processes in Au NP/GaN. The 2PPE intensity varies periodically with the polarization angle, having maxima at θ = 0°, 180°, and 360° (i.e., *p* polarization) and minima at θ = 90° and 270° (i.e., *s* polarization). Similar trends are observed across other kinetic energy ranges. The angular variation in emission intensity closely follows a cos²θ function, where θ is the polarization angle of the pump pulses. This functional form effectively captures the inherent polarization sensitivity of the plasmonic field distributions (Fig. 4c)[64], spectrally characterizing the plasmonic electric field components normal to the Au NP/GaN interface, which are crucial for efficient electron injection. Because the initial momentum distribution of hot electrons is predominantly

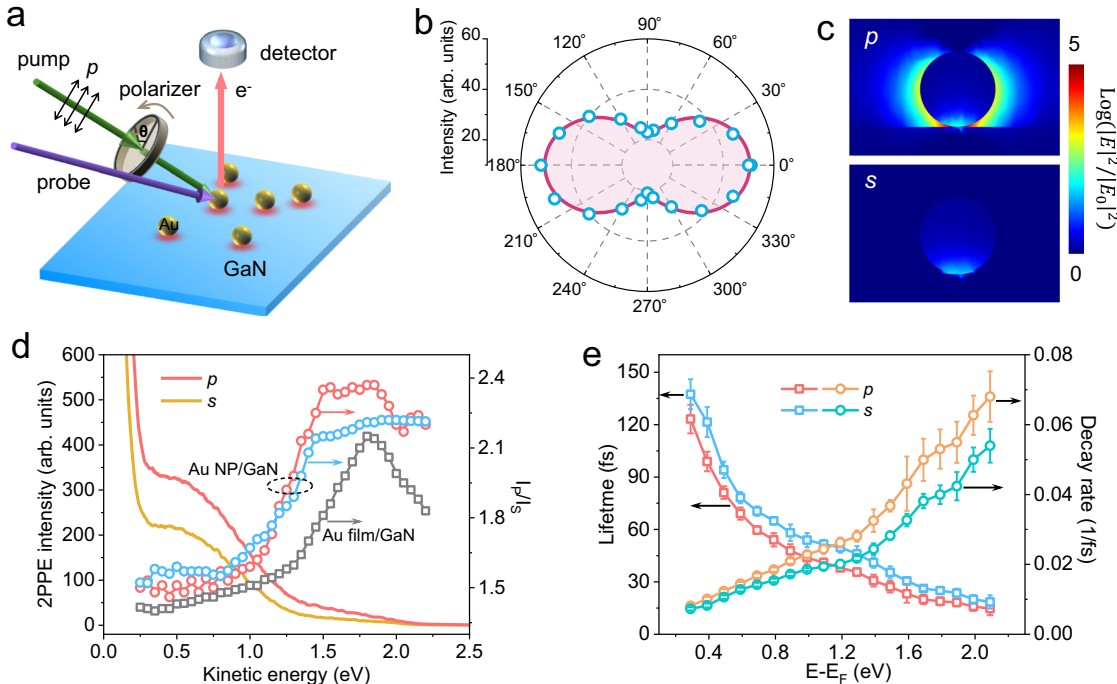

**Fig. 4 | Polarization dependence of charge transfer and hot electron dynamics.**
**a** Schematic illustration showing a polarization-dependent TR-2PPE measurement.
**b** Photoemission intensity (open blue circles) at delay times of 0 fs as a function of pump polarization angle for Au NP/GaN. 0° represents *p* polarization, and 90° represents *s* polarization of the pump. The solid line is a fit to a $\cos^2\theta$ function. **c** The calculated electric field intensity distribution for Au NP/GaN under *p* (top) and *s* (bottom) pump polarization. **d** 2PPE spectra (left axis)) for Au NP/GaN at *p* and *s* pump polarizations, the ratio of *p*- to *s*-polarized photoemission is given on the right axis for both Au NP and -film samples. Experimental data for Au NP/GaN was obtained at pump photon energies of 2.61 eV (open blue circles) and 2.43 eV (open red circles). The open grey squares indicate data for Au film/GaN, excited at 2.43 eV. The probe photon energy is 4.33 eV for all measurements. **e** Hot electron lifetimes (left axis) and decay rates (right axis) as functions of excited electron energy E-E$_F$ for Au NP/GaN at *p* and *s* polarizations, obtained from the fits of time-dependent photoelectron spectra. Error bars in e represent the standard deviations of the fits.

aligned with the laser polarization[10,19], the strong correlation between the 2PPE signal associated with injected electrons and the $\cos^2\theta$ dependence indicates that electron injection efficiency is primarily dictated by the initial trajectory of photoexcited electrons, which undergo minimal momentum relaxation before interfacial injection. This behavior corresponds to a quasi-ballistic transport regime and further corroborates the proposed ultrafast nonthermal electron transfer mechanism.

Figure 4d presents representative TR-2PPE spectra for Au NP/GaN at a delay time of 0 fs, comparing photoemission under *p*- and *s*-polarized pump photons. While the energy range of observed photoelectron emission remains unchanged with polarization, the emission intensities are notably higher under *p*-polarized excitation. To reveal the polarization-dependent energy distribution of excited electrons, we plot the ratio of *p*- to *s*-polarized pump emission (I$_p$/I$_s$) after subtracting time-independent backgrounds taken at 3000 fs for three cases: Au film/GaN at 2.43 eV (grey squares), Au NP/GaN under SPR resonant excitation at 2.43 eV (red circles), and Au NP/GaN in the small detuning regime at 2.61 eV (blue circles). For kinetic energies less than 1.0 eV, the ratio I$_p$/I$_s$ is close to 1.5, agreeing with the expected plasmonic near-field enhancement that determines the excited electrons density (Supplementary Fig. S24). This suggests more efficient photoelectron excitation under *p*-polarized illumination. Notably, for Au NP/GaN, I$_p$/I$_s$ increases with kinetic energies, stabilizing at ~2.3 between 1.5 to 1.9 eV, exceeding the calculated plasmonic near-field enhancement ratio of 1.55. This implies that *p*-polarization preferentially enhances both the generation and collection of high-energy electrons compared to *s*-polarization. Moreover, the maximum I$_p$/I$_s$ for Au NP/GaN is observed under resonant excitation of the SPR due to enhanced plasmonic near-field effect[8]. In contrast, Au film/GaN samples, lacking

a significant surface plasmon effect, only show a pronounced peak in I$_p$/I$_s$ around 1.8 eV arising from interface states[63]. These findings provide direct experimental demonstration that polarization can modulate the energy distribution of hot electrons generated via plasmon excitation, opening an interesting opportunity for engineering ultrafast charge transfer and extraction at interfaces.

Finally, we investigate the influence of pump polarization on hot electron lifetimes. By fitting the temporal evolution of excited electrons under *p*- and *s*-polarized pump excitations (Supplementary Fig. 25), we found that *p*-polarized excitation appears to consistently lead to shorter lifetimes and faster decay rates across the accessible energetic range compared to *s* polarization (Fig. 4e). This contrasts with observations in Au film/GaN without significant charge transfer (Supplementary Fig. 26), where a longer electron lifetime is observed for *p* polarization, as the maximized photon absorption and photoelectron excitation for this configuration significantly extends lifetimes[65]. For Au NP/GaN under *p*-polarized excitation, the facilitated ultrafast electron injection leads to shorter lifetimes for low-energy electrons (E-E$_F$ < 1.10 eV) remaining in Au[23,24,66]. Meanwhile, the reduced lifetimes of high-energy injected electrons (E-E$_F$ > 1.10 eV) are likely associated with enhanced electron transport from the near-interface region into the GaN bulk[53], considering that ultrafast electron dynamics are found to be independent of pump fluences (Supplementary Fig. 22). Future theoretical and experimental studies could be required to further explore polarization-sensitive charge relaxation mechanisms. Nevertheless, our findings are particularly instructive in recognizing the crucial role of polarization in shaping electron energy distributions as well as governing their dynamic transport and relaxation processes, extending beyond previously reported polarization-dependent near-field distributions[63,67]. These insights

open new avenues for refining charge transport models and advancing the design of future devices that rely on polarization-sensitive processes.

In summary, the current study tracking charge transfer process across multiple scales demonstrates ultrafast transfer of nonthermal electron from Au nanoparticles into the GaN conduction band within 40 fs, driven by strong interfacial interactions upon plasmon excitation. This mechanism enables efficient hot electron transport and spatial separation before significant scattering-induced energy losses occur, resulting in a distinct spectral distribution of spatially separated nonthermal electron in GaN. This represents a significant departure from the previously established pathway for efficient charge collection via plasmon-induced interfacial charge-transfer transitions, which required the coupling and mixing of the metal and semiconductor optical states to enable direct charge generation in semiconductors[26,41]. We thus anticipate that our observations may present a universal mechanism for operating plasmonic heterostructures within the true ballistic transport regime by engineering interfacial electronic structures, regardless of the optical properties of the semiconductor, suggesting a versatile framework applicable to a broad plasmonic systems[1]. Furthermore, we uncover a previously unrecognized polarization-dependent effect on energetic distributions, charge separation, and relaxation dynamics in Au NP/GaN systems, providing a new degree of freedom to coherently control the dynamic behavior of hot electrons in plasmonic transport junctions. These findings contribute to our understanding of non-equilibrium electron transfer in plasmonically enhanced heterostructures operating in the ballistic regime, offering significant implications for advancing hot carrier management in plasmonic platforms for future solar energy conversion and optoelectronic applications.

## Methods
### Sample preparation
N-type Ga-polar GaN (0001) wafers were epitaxially grown on sapphire substrates using hydride vapor phase epitaxy (HVPE) (HeFei Crystal Co., Ltd., China). The thickness of the GaN films is 450 nm. The GaN wafers were sequentially rinsed with acetone, ethanol, and deionized water in an ultrasonic bath for 10 min, and then dried with nitrogen gas. Subsequently, the samples were mounted onto a molybdenum sample holder and transferred to an ultrahigh vacuum (UHV) system with a base pressure of approximately $1 \times 10^{-10}$ mbar. For the preparation of Gold/GaN samples, the GaN wafers were transferred to a high vacuum chamber and were initially treated with Ar+ sputtering (400 eV, $2 \times 10^{-4}$ mbar, 5 min) to clean the surfaces. Subsequently, a 2 nm thick Au film was then deposited onto the GaN surface at room temperature using electron beam evaporation at a deposition rate of $1\,\text{Å}\,\text{s}^{-1}$. Finally, the Au film/GaN samples were annealed in UHV at 450 °C for 1 h to form Au nanoparticles and to achieve interfacial contact between Au and GaN. The Au film on GaN without annealing was also used for comparison in some measurements. To prepare the $Al_2O_3$ interlayer between Au and GaN, ~5 nm $Al_2O_3$ films were deposited by thermal ALD at 200 °C using trimethylaluminum (TMA) and $H_2O$ as precursors with $N_2$ as carrier gas. Each ALD cycle comprised a TMA exposure (0.02 s), $N_2$ purge (10 s), $H_2O$ exposure (0.02 s), and $N_2$ purge (10 s). The growth rate was ~1.0 Å per cycle.

### Surface photovoltage microscopy (SPVM)
SPVM measurements were performed using light-modulated Kelvin probe force microscopy (KPFM) with a commercial atomic force microscope (AFM) system (Bruker Dimension Ion V) under ambient conditions. All measurements were conducted using platinum/iridium-coated silicon tips (SCM-PIT) with a spring constant of 3 N/m, tip radius of 25 nm, and a resonance frequency of approximately 75 kHz. In KPFM measurements, the amplitude-modulated mode was used, with a DC voltage of 0.5 V applied at the sample to cause tip oscillation

due to electrostatic interaction between the tip and sample surface. The tip scanning rate was set to 0.5 Hz, and the lift height was fixed at 10 nm to achieve high-quality surface potential imaging while minimizing cross-talk effects from the cantilever as much as possible[36]. The measured surface potential is the difference in work function between the tip and the sample surface. Monochromatic light with a wavelength of 520 nm (5 mW/cm$^2$), provided by a xenon lamp (Beijing Perfectlight) equipped with a monochromator (Zolix Omni-λ 500), was used to excite surface plasmons of the Au nanoparticles for irradiated-KPFM measurements. SPV mapping is obtained by subtracting the surface potential image in dark conditions from the surface potential image under illumination, which helps directly visualize the spatial distribution of photogenerated charges on the sample surface. All imaging data were analyzed using NanoScope Analysis software.

### Time-resolved two-photon photoemission spectroscopy (TR-2PPE)
TR-2PPE was employed to monitor the occupation dynamics of surface electronic states and the dynamics of electrons that are photoexcited from occupied states within the valence band or band gap to virtual intermediate states and, from there relax through near-surface states towards the Fermi level. TR-2PPE experiments were carried out in UHV, with a base pressure of around $1.3 \times 10^{-10}$ mbar at room temperature (20.6 °C). To generate the pulses used for photoexcitation and emission, the laser pulses of a Coherent Vitara Titanium Sapphire (Ti:Sa) oscillator are first passed through a grating-based compressor/expander unit and subsequently amplified by a Coherent RegA 9050 amplifier, which is pumped by a Verdi 12 laser. The resulting 800 nm pulses have a full width at half maximum (FWHM) of approximately 50 fs and 150 kHz repetition rate with pulse energies of 10 μJ. In TR-2PPE, the sample is first photoexcited by a pump pulse to produce electronic excitation, after which a time-delayed probe pulse emits the photoexcited electrons through a two-photon photoemission process (as schematically shown in Supplementary Fig. 14). To generate pump and probe pulses of adjustable photon energies, part of the 800 nm light is frequency doubled to 400 nm using second harmonic generation in a β-barium borate (BBO) crystal, which then pumps two non-collinear parametric amplifiers (NOPA). One NOPA generates the visible (VIS) pump beam, with tunable photon energies from 475 nm (2.61 eV) to 560 nm (2.21 eV) at pulse energies of around 80 nJ directly after the NOPA. The second NOPA is tuned to wavelengths ranging from 520 nm (2.38 eV) to 600 nm (2.06 eV), with pulse energies of about 220 nJ. This pulse is subsequently frequency-doubled to produce the UV probe beam, which itself is adjustable from 265 nm (4.68 eV) to 330 nm (3.76 eV), at a pulse energy of around 1.4 nJ. The pump and probe pulses are each chirp-compensated using a pair of quartz prisms, resulting in FWHM of around 40 fs at the sample. For experiments on pure GaN, two UV pulses were used in a pump-probe configuration by splitting the UV beam with a 2:1 pulse energy split between pump and probe (see schematic in Supplementary Fig. 9a). The overlap of the VIS and UV beam spots on the sample is about $100 \times 100\ \mu m^2$. VIS pump pulse energies can be varied within 3–25 μJ using neutral density filters. A time-of-flight (TOF) detector with a 7.3° acceptance angle and approximately 50 meV energy resolution is positioned 3-5 mm from the sample surface to detect the photo-emitted electrons. This distance is selected to optimize angular acceptance while minimizing the detection of secondary (scattered) electrons. The pump and probe beams are incident at approximately 45° relative to the surface normal. To prevent damage to the detector, the count rate of photo-generated electrons is maintained below $10^{17}\ cm^{-3}\ s^{-1}$ by adjusting the pump and probe pulse energies. Autocorrelation (AC, ~35 fs) and cross-correlation (CC, ~40 fs) of pump and probe pulses were conducted using a Cu (111) single crystal with an occupied surface state, resulting from the emission via virtual intermediate states in the sp-band gap[39]. Prior to AC and CC measurements,

the Cu sample was cleaned using multiple sputtering cycles ($Ar^+$, 650 eV, $1.5 \times 10^{-5}$ mbar), each followed by annealing at 527 °C for 20 min. In post-processing, to obtain the time-dependent TR-2PPE signal, the TR-2PPE spectra were corrected by subtracting the background signal taken at around 3 ps pump-probe delay, where no more time-dependent emission was observed.

## Conductive atomic force microscopy (CAFM)
The solid-state current-voltage (I-V) characteristic was measured using CAFM with a commercial AFM system (Bruker Dimension Ion V), which operates under ambient conditions (temperature, 22 °C; humidity, 30%). In this experiment, the AFM was operated in PeakForce TUNA mode using an SCM-PIT (Bruker) tip to collect conductive currents. This approach enables high-sensitivity and high-resolution current measurements by providing precise force control and eliminating lateral forces. Initially, the surface topography of the Au NP/GaN sample was imaged over a 2 μm × 2 μm area to identify the positions of Au nanoparticles. Subsequently, the conductive tip was positioned centrally on an Au nanoparticle, and the I–V curve was recorded by sweeping the sample bias from −5 V to +5 V in ramp mode. To quantify the Schottky barrier height of the interface, the measured I-V curve was fitted using thermionic emission simulation as follows:

$$I = AA^* T^2 e^{\left(-\frac{q\varphi_B}{kT}\right)} e^{\left[\frac{q(V-IR)}{nkT}\right]} \tag{1}$$

where $A = \pi r_{tip}^2$ is the tip contact area with sample surface, $A^* = 110$ A cm$^{-2}$ K$^{-2}$ is the Richardson constant, $V$ is the applied voltage on tip, $\varphi_B$ is the Schottky barrier height at interface, $q$ is the electron charge, $k$ is the Boltzmann constant, $T$ is the absolute temperature, and $n$ is the ideality factor.

## Ultraviolet–visible (UV-Vis) absorption spectroscopy
UV-Vis absorption spectroscopy was performed using a PerkinElmer Lambda 950 spectrophotometer to characterize the optical properties of the samples. This instrument is equipped with a dual-beam optical system and a high-performance photomultiplier tube (PMT) for the UV-Vis range as well as an indium gallium arsenide (InGaAs) detector for the NIR range, providing high accuracy across a broad spectral range from 190 to 3300 nm.

## X-ray photoelectron spectroscopy (XPS)
XPS measurements were performed using a SPECS Focus 500 monochromator and a Phoibos 100 electron energy analyzer. An aluminum K-alpha X-ray (1486.74 eV) operated at 300 W, was used as the X-ray source. Measurements were conducted at a photoelectron take-off angle of 90°, an X-ray source to analyzer angle of 54.7°, and a pass energy of 10 eV. Data was collected in energy steps of 0.05 eV for high-resolution core level peaks and was post-processed using Shirley background subtraction.

## Ultraviolet Photoelectron Spectroscopy (UPS)
UPS was employed to assess the surface work functions and resolve the valence band features of the samples. All UPS measurements were conducted at room temperature using the same apparatus as the XPS measurements. Helium I (He-I) radiation with a photon energy of 21.2 eV was utilized as the UV light source, with the lamp operating at 35 W. The incident light was oriented at a 50° angle relative to the surface normal, while photoelectrons were detected along the surface normal direction. An overpotential of −5.0 V was applied between the sample and the detector to distinguish the secondary electron cutoff.

## Raman spectroscopy
Raman spectroscopy was performed using a Renishaw Raman microscope equipped with an Olympus ×100 objective. A continuous-wave 532 nm excitation wavelength was used for excitation. The laser was focused onto the sample using an optical microscope, which facilitated the collection of the scattered light for analysis by a spectrometer. The Raman spectra were calibrated using the 520 cm$^{-1}$ peak from a silicon wafer before each experiment.

## Numerical simulations
COMSOL Multiphysics was utilized to simulate the electromagnetic field distribution and absorption spectrum of the Au/GaN heterostructure under varying polarization conditions. The simulation model consisted of a spherical Au nanoparticle with a diameter of 5 nm positioned on a GaN substrate (refractive index = 2.45) within a three-dimensional computational domain surrounded by air. A plane wave source was applied, propagating at an angle of 45° relative to the z-axis, with two polarization configurations: p-polarized light, where the electric field was in the xz-plane, and s-polarized light, where the electric field aligned along the x-axis. Perfectly matched layers (PMLs) were added to the outer edges of the computational domain to absorb outgoing waves and eliminate reflections at the domain boundaries. A mesh size of 2 nm was implemented for all simulations. The electromagnetic field distribution simulations were performed at the plasmon resonance wavelength of 520 nm. To simulate the absorption spectrum, the absorbed power was integrated over the nanoparticle surface for each wavelength, and the absorption cross-section was determined under both polarization conditions.

## Data availability
The data that support the findings of this study are available from the corresponding authors upon request.

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

## Acknowledgements

We acknowledge the financial support from National Program on Key Basic Research Project (2024YFA1210802, Y.G., and 2021YFA1500600, F.F.), Fundamental Research Center of Artificial Photosynthesis (FReCAP, C.L., F.F.), the National Natural Science Foundation of China (22325205, F.F., 22088102, F.F. and 22472170, Y.G.), CAS Projects for Young Scientists in Basic Research (YSBR-004, F.F.), Fundamental Research Funds for the Central Universities (20720220011, F.F.), New Cornerstone Science Foundation through the XPLORER PRIZE (F.F.), and German Research Foundation (DFG project PAK 981, project no. FR 4025/2-1, D.F., KR4816/1-1, R.K).

## Author contributions

Y.G., F.F. and C.L. conceived the research idea and designed the project. Y.G. performed 2PPE spectroscopy, analysed the experimental data, and wrote the manuscript. J.D. and D.F. helped with the experimental setup. K.H. prepared the plasmonic nanostructure, and C.H. conducted the characterization of XPS and UPS. Y.X. carried out the SPV measurements. Q.Z. performed the COMSOL simulations. J.D. built the analytical model for calculating the electron relaxation, participated in analyzing the results, and co-wrote the paper. D.F. and R.K. discussed the data and revised the manuscript.

## Funding

## Competing interests

The authors declare no competing interests.
