## [Transparent Peer Review file · Nature Communications]

Ultrafast nonthermal electron transfer at plasmonic interfaces

Corresponding Author: Dr Dennis Friedrich

Version 0:

Reviewer comments:

Reviewer #1

(Remarks to the Author)

The authors have studied the hot electrons transfer dynamics in the Au NP/GaN heterodimers. They have claimed that the ballistic transfer mechanism plays the crucial role and dependence of the injected hot electron population on pump light polarization were studied. These results were verified by solid experimental evidences. However, it is overall an incremental work with only finite advances to the field. I can not recommend this work to be published on Nature Communications based on the following considerations listed below.

1. The ballistic transfer mechanism has been proved by many experimental studies and is indeed belong to the indirect electron mechanism. The physical understandings that indirect electron transfer lifetime beyond 100 fs are highly doubtful (Ref 1 in the manuscript), because more than two electron-electron scatterings would severely lower down the hot electron energy below the Schottky barrier.
2. Direct hot electron transfer is assigned to the interfacial electron transfer that hot electrons are generated at interface. Polarization dependent hot electron transfer is in line with the polarization dependent hot electron direction generated by surface damping of SPR. These viewpoints have been explained thoroughly in Khurgin, Nanophotonics 2020, doi: 10.1515/nanoph-2019-0396.

Reviewer #2

(Remarks to the Author)

The authors present a detailed study of ultrafast hot electron injection from Au nanoparticles into GaN using TR-2PPE and SPVM. The overall experimental approach is strong, and the combination of time-resolved measurements with surface-sensitive techniques provides convincing evidence for fast electron transfer occurring within 40 fs. The authors carefully compared Au NP/GaN with Au film/GaN and demonstrated how polarization of the pump pulse strongly influences hot electron generation, injection efficiency, and decay dynamics. Their observation that p-polarized excitation enhances plasmonic hot electron production and accelerates interfacial electron injection is an important insight into how optical control parameters affect ultrafast charge transfer. We fully support the publication of this manuscript pending the following suggestions to further improve this excellent work.

Minor Comments:

1. The authors used 7 nm Au nanoparticles, but did not discuss how variations in nanoparticle size or shape might influence the electron transfer process. Could changes in size affect the injection efficiency, speed, or even the Schottky barrier height?
2. The authors estimate a 31% injection efficiency. How consistent is this value across repeated experiments or under varying pump fluences?
3. How stable is the Au NP/GaN interface during repeated laser pulses or long-term measurements? Does the charge injection efficiency change over time?
4. The authors suggest that electrons are injected into GaN and then back-injected into the Au NP on a ~120 fs timescale. However, earlier they mention that hot electrons thermalize in GaN within ~18 fs through LO phonon scattering. It's not obvious how the electrons could remain available for back-injection after they should have already relaxed. The authors

should clarify whether some of the injected electrons remain near the interface unthermalized, or if another mechanism is responsible for the observed lifetime extension.

5. In their TR-2PPE interpretation, the authors assign all sub-barrier photoemission ($E < 1.1$ eV) to thermalized electrons in the Au NP, but during early times (~ 0 –50 fs), it is possible that relaxed injected electrons in GaN may also contribute signal within this energy range. Without spectral deconvolution or additional modeling, the sharp energy-based separation may oversimplify the actual contributions to the measured signal. The authors should discuss such simplification.

6. The authors should discuss why they observe stronger AuNP/GaN interaction than AuFilm/GaN interactions. It seems counterintuitive that colloidal AuNPs adsorbed onto the GaN would form a stronger interaction than a PVD grown film. Could the AuNP ligands be responsible for these differences?

7. The authors mention that for some measurements the gold film was annealed, forming gold nanoislands. This would result in a system more like the AuNP/GaN sample and should exhibit similar results. The authors should describe these comparisons and explain the results.

8. The authors exclude contributions from Auger recombination based on a lack of a pump power dependence. However, the effect of Auger scattering is readily seen in TR-2PPE literature showing a suppression of the d-band effect in the hot electron lifetime (doi:10.1103/PhysRevB.57.12812 and 10.1016/S0039-6028(96)01353-2). So, it is odd to exclude such a mechanism from hot carrier generation and injection, especially at excitation wavelengths below the d-band onset. Also, Auger recombination results in the same athermal energy distribution as intraband excitation and given the excitation energy used interband excitation should be the dominant absorption mechanism.

9. It is surprising that the hot electrons injected into the GaN are so short lived. IR-transient absorption spectra of hot holes injected into p-type GaN exhibit injected carrier lifetimes of tens of picoseconds (10.1038/s41563-020-0737-1). While the authors explain the short lifetime is due to optical phonons, shouldn't one expect similar reasoning for injected hot holes?

Major Comments

1. While the gold film/GaN heterostructure seems to lack a strong injection dependence, the authors should have a control experiment of the gold nanoparticles/film on a material that lacks interfacial charge transfer such as SiO₂ or Al₂O₃ to compare the dynamics against. It is surprising that such a control was not included.

Reviewer #3

(Remarks to the Author)

Version 1:

Reviewer comments:

Reviewer #1

(Remarks to the Author)

The authors have clearly claimed all my comments and I would like to recommend its publication on Nature Communications as its current version.

Reviewer #2

(Remarks to the Author)

The Authors have sufficiently addressed the reviewers' comments.

REVIEWER COMMENTS

Reviewer #1 (Remarks to the Author):

The authors have studied the hot electrons transfer dynamics in the Au NP/GaN heterodimers. They have claimed that the ballistic transfer mechanism plays the crucial role and dependence of the injected hot electron population on pump light polarization were studied. These results were verified by solid experimental evidences. However, it is overall an incremental work with only finite advances to the field. I can not recommend this work to be published on Nature Communications based on the following considerations listed below.

1. The ballistic transfer mechanism has been proved by many experimental studies and is indeed belong to the indirect electron mechanism. The physical understandings that indirect electron transfer lifetime beyond 100 fs are highly doubtful (Ref 1 in the manuscript), because more than two electron-electron scatterings would severely lower down the hot electron energy below the Schottky barrier.

2. Direct hot electron transfer is assigned to the interfacial electron transfer that hot electrons are generated at interface. Polarization dependent hot electron transfer is in line with the polarization dependent hot electron direction generated by surface damping of SPR. These viewpoints have been explained thoroughly in Khurgin, Nanophotonics 2020, doi: 10.1515/nanoph-2019-0396.

Reviewer #2 (Remarks to the Author):

The authors present a detailed study of ultrafast hot electron injection from Au nanoparticles into GaN using TR-2PPE and SPVM. The overall experimental approach is strong, and the combination of time-resolved measurements with surface-sensitive techniques provides convincing evidence for fast electron transfer occurring within 40 fs. The authors carefully compared Au NP/GaN with Au film/GaN and demonstrated how polarization of the pump pulse strongly influences hot electron generation, injection efficiency, and decay dynamics. Their observation that p-polarized excitation enhances plasmonic hot electron production and accelerates interfacial electron injection is an important insight into how optical control parameters affect ultrafast charge transfer. We fully support the publication of this manuscript pending the following suggestions to further improve this excellent work.

Minor Comments:

1. The authors used 7 nm Au nanoparticles, but did not discuss how variations in nanoparticle size or shape might influence the electron transfer process. Could changes in size affect the injection efficiency, speed, or even the Schottky barrier height?
2. The authors estimate a 31% injection efficiency. How consistent is this value across repeated experiments or under varying pump fluences?
3. How stable is the Au NP/GaN interface during repeated laser pulses or long-term measurements? Does the charge injection efficiency change over time?
4. The authors suggest that electrons are injected into GaN and then back-injected into the Au NP on a ~ 120 fs timescale. However, earlier they mention that hot electrons thermalize in GaN within ~ 18 fs through LO phonon scattering. It's not obvious how the electrons could remain available for back-injection after they should have already relaxed. The authors should clarify whether some of the injected electrons remain near the interface unthermalized, or if another mechanism is responsible for the observed lifetime extension.
5. In their TR-2PPE interpretation, the authors assign all sub-barrier photoemission ($E < 1.1$ eV) to thermalized electrons in the Au NP, but during early times (~ 0 –50 fs), it is possible that relaxed injected electrons in GaN may also contribute signal within this energy range. Without spectral deconvolution or additional modeling, the sharp energy-based separation may oversimplify the actual contributions to the measured signal. The authors should discuss such simplification.
6. The authors should discuss why they observe stronger AuNP/GaN interaction than AuFilm/GaN interactions. It seems counterintuitive that colloidal AuNPs adsorbed onto the GaN would form a stronger interaction than a PVD grown film. Could the AuNP ligands be responsible for these differences?
7. The authors mention that for some measurements the gold film was annealed, forming gold nanoislands. This would result in a system more like the AuNP/GaN sample and should exhibit similar results. The authors should describe these comparisons and explain the results.
8. The authors exclude contributions from Auger recombination based on a lack of a pump power dependence. However, the effect of Auger scattering is readily seen in TR-2PPE literature showing a suppression of the d-band effect in the hot electron lifetime (doi:10.1103/PhysRevB.57.12812 and 10.1016/S0039-6028(96)01353-2). So, it is odd to exclude such a mechanism from hot carrier generation and injection, especially at excitation wavelengths below the d-band onset. Also, Auger recombination results in the same athermal energy distribution as intraband excitation and given the excitation energy used interband excitation should be the dominant absorption mechanism.
9. It is surprising that the hot electrons injected into the GaN are so short lived. IR-transient absorption spectra of hot holes injected into p-type GaN exhibit injected carrier lifetimes of tens of picoseconds (10.1038/s41563-020-0737-1). While the authors explain the short lifetime is due to optical phonons, shouldn't one expect similar reasoning for injected hot holes?

Major Comments

1. While the gold film/GaN heterostructure seems to lack a strong injection dependence, the authors should have a control experiment of the gold nanoparticles/film on a material that lacks interfacial charge transfer such as SiO₂ or Al₂O₃ to compare the dynamics against. It is surprising that such a control was not included.

Reviewer #3 (Remarks to the Author):

Responses to the reviewer(s)' comments

We would like to express our sincere appreciation to reviewers for reviewing our manuscript entitled " **Ultrafast nonthermal electron transfer at plasmonic interfaces**" (NCOMMS-25-17572-T). We have thoroughly revised our manuscript in accordance with the reviewers' comments, incorporated additional experimental evidence, and provide here a point-by-point response to all the comments.

Reviewer #1:

(Remarks to the Author): The authors have studied the hot electrons transfer dynamics in the Au NP/GaN heterodimers. They have claimed that the ballistic transfer mechanism plays the crucial role and dependence of the injected hot electron population on pump light polarization were studied. These results were verified by solid experimental evidences. However, it is overall an incremental work with only finite advances to the field. I can not recommend this work to be published on Nature Communications based on the following considerations listed below.

Response: Thank you for recognizing the solid experimental evidence supporting our results. Your comments have been carefully considered, and we provide detailed, point-by-point responses below. We hope that these clarifications demonstrate the advancement and novelty of our study.

Regarding the concern that our study represents only incremental advances, we respectfully emphasize that the novelty of our work lies in tracking plasmonic hot electron transfer dynamics across spatial, temporal, and energy domains that previous studies have not been able

to access. By integrating femtosecond time-resolved two-photon photoemission (TR-2PPE) with nanometer-resolved surface photovoltage microscopy (SPVM), we directly reveal the ballistic injection of nonthermal electrons from Au nanoparticles into the GaN conduction band within 40 fs, occurring prior to significant scattering-induced energy loss. This combined approach enabled us to disentangle nonthermal electron transfer from conventional thermalized pathways, offering quantitative, energy-resolved mechanistic insights that were previously unavailable. Furthermore, our polarization-resolved measurements uncovered a new degree of coherent control over hot electron energy distributions and relaxation dynamics under nonthermal transport conditions. Taken together, these methodological advances provide a versatile experimental framework for probing and engineering plasmonic heterostructures in the true ballistic regime, delivering meaningful progress that significantly advances our understanding of plasmonic hot electron dynamics.

1. The ballistic transfer mechanism has been proved by many experimental studies and is indeed belong to the indirect electron mechanism. The physical understandings that indirect electron transfer lifetime beyond 100 fs are highly doubtful (Ref 1 in the manuscript), because more than two electron-electron scatterings would severely lower down the hot electron energy below the Schottky barrier.

Response: We thank the reviewer for the comment and would like to clarify the relationship between ballistic and indirect electron transfer mechanisms, as well as the concern about transfer timescale exceeding 100 fs. In the literature, electron injection from a metal nanoparticle into a semiconductor is broadly described as “indirect” (*Nat. Rev. Methods Primers* **2023**, 3, 12; *Nat. Rev. Chem.* **2022**, 6, 259–274), and under this framework, ballistic transfer may be considered as a particular case of indirect charge transfer. Nevertheless, it is essential to emphasize that ballistic transfer is mechanistically distinct from the conventional indirect (thermalized) process in timescale, energy domain, and scattering processes. In the

conventional indirect mechanism, electrons undergo multiple electron–electron and electron–phonon scattering events, leading to partial thermalization, significant energy loss, and transfer times typically longer than 100 fs (*Sci. Adv.* **2024**, *10*, eadp3353; *Annu. Rev. Phys. Chem.* **2017**, *68*, 379–398), which indeed limits their ability to overcome interfacial barriers such as the Schottky barrier. In contrast, ballistic electron transfer occurs on ultrafast timescales (<40 fs in our Au NP/GaN system, ~11 fs in Au/TiO₂ reported in ref. 1), prior to significant scattering, thereby enabling nonthermal electrons to retain sufficient energy for efficient interfacial injection. Our TR-2PPE and nanometer-resolved SPVM measurements provide direct experimental evidence of injection within ~40 fs, together with a distinct nonthermal energy distribution, thereby substantiating the ballistic character of the process. Therefore, while ballistic transfer can be formally classified within the broad framework of indirect electron transfer, it represents an ultrafast, nonthermal channel that is fundamentally different from the slower, energy-relaxed indirect mechanism, and our observations are fully consistent with the expected physical picture.

We have revised the relevant discussion to clarify this point in the revised manuscript as below:

(P11) “Moreover, conventional indirect charge transfer is unlikely to produce a nonthermal distribution in GaN, as the electron–electron scattering events involved in this process would thermalize the electron distribution⁴¹. Therefore, the preservation of a nonthermal electron distribution in GaN up to the pump energy at 0 fs (Fig. 2d) strongly suggests ultrafast nonthermal electron transfer occurring within the pulse duration (~40 fs), before significant e–e scattering dissipates their energy. This ultrafast transfer occurs on a timescale beyond the resolution of our measurement, consistent with recent observations of charge injection in Au/TiO₂ on a timescale of ~11 fs determined by terahertz emission¹. Thus, the nonthermal electron transfer demonstrated here is distinct from the conventional indirect mechanism,

which generally involves scattering-induced thermalization and slower energy-relaxed transfer¹⁸.”

2. Direct hot electron transfer is assigned to the interfacial electron transfer that hot electrons are generated at interface. Polarization dependent hot electron transfer is in line with the polarization dependent hot electron direction generated by surface damping of SPR. These viewpoints have been explained thoroughly in Khurgin, *Nanophotonics* 2020, doi: 10.1515/nanoph-2019-0396.

Response: We sincerely thank the reviewer for the insightful comments and fully acknowledge the comprehensive theoretical framework for hot electron transfer mechanisms presented in the review by Khurgin (*Nanophotonics* 2020, doi:10.1515/nanoph-2019-0396). While our work is connected to this theoretical study, it provides significant extensions in several aspects.

First, we provide direct experimental evidence of ultrafast nonthermal electron transfer at the Au NP/GaN interface with femtosecond temporal resolution and nanometer spatial resolution, enabling us to disentangle the ballistic transfer channel from the slower, energy-relaxed indirect processes. To the best of our knowledge, such simultaneous tracking electron behavior across temporal, spatial, and energy domains has not been reported previously, allowing quantitative, energy-resolved insights into interfacial hot electron dynamics.

Second, our polarization-resolved measurements reveal a previously unrecognized degree of coherent control over the energetic distribution, spatial separation, and relaxation dynamics of hot electrons. These observations cannot be fully captured by polarization-related theoretical models, including the review by Khurgin, as computational approaches face difficulties in accurately representing the complex metal/semiconductor interface and simulating energy-resolved ultrafast dynamics.

Third, we demonstrate that this ballistic transfer occurs within ~ 40 fs, prior to significant electron-electron scattering, thereby preserving sufficient electron energy to efficiently cross the Schottky barrier, a feature that previous theoretical treatments could not directly verify.

Therefore, these results establish a mechanistically distinct, experimentally validated regime of hot electron transfer, provide new insights into non-equilibrium plasmonic dynamics, and offer a versatile framework for controlling hot carrier behavior in plasmonic heterostructures, going well beyond an incremental extension of existing knowledge.

Review #2:

(Remarks to the Author): The authors present a detailed study of ultrafast hot electron injection from Au nanoparticles into GaN using TR-2PPE and SPVM. The overall experimental approach is strong, and the combination of time-resolved measurements with surface-sensitive techniques provides convincing evidence for fast electron transfer occurring within 40 fs. The authors carefully compared Au NP/GaN with Au film/GaN and demonstrated how polarization of the pump pulse strongly influences hot electron generation, injection efficiency, and decay dynamics. Their observation that p-polarized excitation enhances plasmonic hot electron production and accelerates interfacial electron injection is an important insight into how optical control parameters affect ultrafast charge transfer. We fully support the publication of this manuscript pending the following suggestions to further improve this excellent work.

Response: We thank the reviewer for the highly positive remarks and valuable guidance on our manuscript.

Minor Comments:

1. The authors used 7 nm Au nanoparticles, but did not discuss how variations in nanoparticle size or shape might influence the electron transfer process. Could changes in size affect the injection efficiency, speed, or even the Schottky barrier height?

Response: We thank the reviewer for raising this important point. We agree that nanoparticle size and shape can in principle influence hot-carrier generation and electron transfer dynamics. In this work, we deliberately fixed the nanoparticle size at ~7 nm to minimize additional variables and to focus on the interfacial physics under well-controlled conditions. Our choice of 7 nm was also motivated by physical considerations: if the particles are too small (<3 nm), the surface plasmon resonance (SPR) effect becomes less pronounced (*Phys. Rev. Lett.* **2021**, *126*, 173902), limiting efficient hot-carrier generation; conversely, if the particles are too large, probing photoelectrons from the Au/GaN interface becomes challenging, as the photoelectron escape depth in Au at the photon energy (4.49 eV) employed in this study is only about 11 nm (*Phys. Rev. B* **1998**, *58*, 10948-10952). Thus, 7 nm represents a balance between supporting a clear plasmonic response and ensuring that interfacial electronic processes can still be effectively detected.

While we did not systematically vary the particle size in this work, previous reports have demonstrated that nanoparticle size can influence electron injection efficiency (*Nano Lett.* **2025**, *25*, 3253–3258; *ACS Photonics* **2024**, *11*, 2255–2262), apparent Schottky barrier height (*Nano Lett.* **2015**, *15*, 51–55; *Nanoscale*, **2018**, *10*, 22180–22188), and injection process (*Chem. Phys. Lett.*, **2021**, *770*, 138457; *Nano Lett.* **2025**, *25*, 3253–3258). Nevertheless, variations in metal nanoparticle size do not affect our main conclusion that ultrafast nonthermal electron transfer occurs in our system, as this mechanism is primarily governed by interfacial interactions. Given the complex influence of nanoparticle size on multiple aspects of the electron transfer process,

we believe that a systematic experimental investigation of size/shape effects would be valuable and plan to pursue this in future work.

We have added a paragraph in the revised manuscript to clarify this point:

(P5): “The size and morphology of metal nanoparticles are known to influence the efficiency of hot-carrier injection, the Schottky barrier height, and the injection process. In this study, we selected ~7 nm Au nanoparticles because this size ensures sufficiently strong plasmonic excitation while remaining within the electron escape depth (~11 nm) for the subsequent 2PPE studies, thereby allowing us to directly probe the ultrafast interfacial charge transfer dynamics at the Au/GaN interface.”

2. The authors estimate a 31% injection efficiency. How consistent is this value across repeated experiments or under varying pump fluences?

Response: The estimated ~31% injection efficiency represents the value obtained under our standard experimental conditions. In repeated measurements under the same pump fluence, we observe variations within $\pm 3\%$, indicating good reproducibility. We also examined TR-2PPE spectra of Au NPs/GaN under different pump fluences, as shown in Figure R1. While the pump fluence influences the overall photoemission intensity, it has a negligible impact on the hot-electron energy distribution. From the energy-resolved spectra at 0 fs, we estimate that within the experimental fluence range the injection efficiency remains essentially constant at $31\% \pm 5\%$. These results confirm that the injection efficiency is highly reproducible and only weakly dependent on excitation fluence.

Figure R1. (a-c) Representative pseudo-color plot of Au NP/GaN excited at a photon energy of 2.58 eV under pump fluences of (a) 10 nJ/cm², (b) 26 nJ/cm², and (c) 44 nJ/cm². (d-f) Corresponding 2PPE spectra of Au NP/GaN sample measured at pump fluences of (d) 10 nJ/cm², (e) 26 nJ/cm², and (f) 44 nJ/cm².

3. How stable is the Au NP/GaN interface during repeated laser pulses or long-term measurements? Does the charge injection efficiency change over time?

Response: We thank the reviewer for this important question. The TR-2PPE spectra remained highly reproducible across repeated scans (Figure R2), demonstrating that the Au NP/GaN interface is stable under our experimental conditions and that the estimated charge-injection efficiency does not exhibit systematic variations over time. To further assess stability, we monitored the photoemission signal over repeated pulse periods (Figure R3). The photoemission traces are reproducible across different pulse cycles, and after 20 consecutive pulse cycles (total measurement time is about 3 h) we observed no measurable decline in signal intensity nor any change in spectral features. These observations demonstrate that, under the present excitation conditions, the Au NP/GaN interface remains robust and no photoinduced

degradation was detected within the tested time window. The relevant discussion and corresponding data have been included in the revised manuscript and Supplementary Information (Figure S12c) as below:

(P9) “The TR-2PPE spectra of Au NP/GaN were highly reproducible over repeated pulse cycles (Supplementary Fig. 12c), with no detectable decrease in intensity or change in spectral features, confirming the robustness of the interface under prolonged excitation.”

Figure R2. 2PPE spectra of the Au NP/GaN sample measured in the repeated experiments. The excitation conditions were identical across all measurements.

Figure R3. Time-dependent 2PPE signal for Au NP/GaN excited at a pump energy of 2.34 eV over repeated pulse cycles, suggesting the robustness of the interface under prolonged excitation. The 2PPE signal was obtained at a kinetic energy of 2.0 eV.

4. The authors suggest that electrons are injected into GaN and then back-injected into the Au NP on a ~120 fs timescale. However, earlier they mention that hot electrons thermalize in GaN within ~18 fs through LO phonon scattering. It's not obvious how the electrons could remain available for back-injection after they should have already relaxed. The authors should clarify whether some of the injected electrons remain near the interface unthermalized, or if another mechanism is responsible for the observed lifetime extension.

Response: We thank the reviewer for this insightful comment. Under our experimental conditions, hot electrons in the Au nanoparticles with energies of $1.1 \text{ eV} \leq E - E_F \leq 2.34 \text{ eV}$ are capable of being injected into the GaN conduction band. After injection, these hot electrons scatter toward the conduction band minimum (CBM) of GaN through LO phonon–electron scattering. Although the e–ph scattering time in GaN is fast (~18 fs), electrons that have not fully relaxed to the CBM after multiple scattering events may remain available for back-injection into Au on a longer timescale (~120 fs). Similar behavior has been reported in the Au/WSe₂ system (*Adv. Mater.* **2023**, *35*, 2209100). We also fully agree with the reviewer's suggestion that unthermalized electrons near the interface could contribute to this back-injection process. However, our energy-dependent lifetime analysis did not reveal discernible features attributable to interface states (Figure 3c), suggesting that their contribution to back-injection is minor within the experimental uncertainty.

We have added the relevant discussion to clarify this point in the Supplementary Information, as shown below:

(P8-P9) “Hot electrons in Au nanoparticles with energies of 1.1–2.34 eV above E_F can be injected into the GaN conduction band. After injection, they relax toward the CBM via LO phonon scattering. While the scattering time in GaN is fast (~18 fs), electrons that have not

fully relaxed after multiple scattering events may back-inject into Au on a longer timescale (~120 fs), leading to delayed thermalization of lower-energy carriers. The absence of discernible features associated with interface states in the energy-dependent lifetime analysis further suggests that their contribution to back-injection is negligible within experimental uncertainty.”

5. In their TR-2PPE interpretation, the authors assign all sub-barrier photoemission ($E < 1.1$ eV) to thermalized electrons in the Au NP, but during early times (~0–50 fs), it is possible that relaxed injected electrons in GaN may also contribute signal within this energy range. Without spectral deconvolution or additional modeling, the sharp energy-based separation may oversimplify the actual contributions to the measured signal. The authors should discuss such simplification.

Response: In our TR-2PPE measurements, the sub-barrier signal ($E < 1.1$ eV, blue region in Figure R4) arises from thermalized electrons in Au, whereas high-energy electrons (green region in Figure R4) injected into the GaN conduction band relax at energies outside this spectral window and therefore do not contribute to the sub-barrier signal. The use of an energy-based separation provides a clear framework to distinguish the dominant contributions from Au thermalized carriers versus high-energy electrons injected into GaN. We recognize that such a sharp separation may be an oversimplification for quantitative analysis and that more rigorous spectral deconvolution would be valuable. Nevertheless, the robustness of the energy- and time-dependent trends across the spectra supports our assignment, and our central conclusion regarding ultrafast nonthermal charge transfer remains unaffected.

Figure R4. Schematic illustration of TR-2PPE process for Au NP/GaN under plasmon excitation. The low-energy portion (blue region) of the 2PPE spectrum arises from low-energy electrons that remain in Au without being injected, whereas high-energy electrons injected into GaN contribute to the high kinetic energy region (green region) of the spectrum. Even after thermalization to the GaN conduction band minimum, their kinetic energies remain above 1.1 eV. Therefore, the injected electrons in GaN do not contribute to the low-energy region.

6. The authors should discuss why they observe stronger AuNP/GaN interaction than AuFilm/GaN interactions. It seems counterintuitive that colloidal AuNPs adsorbed onto the GaN would form a stronger interaction than a PVD grown film. Could the AuNP ligands be responsible for these differences?

Response: The Au nanoparticles in our study were obtained by high-temperature annealing of deposited Au thin films, rather than being colloidal Au NPs. Therefore, surface ligands are not present and do not influence the interface. Compared with the Au film/GaN samples, the stronger interaction observed for Au NP/GaN arises from the high-temperature annealing

process, which promotes interfacial atomic diffusion and enhances adhesion with the underlying GaN. Such enhancement of interface interactions through nanoparticle formation by annealing has been widely reported in various systems (*Angew. Chem. Int. Ed.* **2024**, *63*, e202402435; *Nat. Nanotechnol.* **2020**, *19*, 1312-1318; *Nat. Nanotechnol.* **2018**, *13*, 953-958).

We have revised the relevant discussion to clarify this point in the revised manuscript, as shown below:

(P8) “This is indicative of strong interfacial interactions in Au NP/GaN compared to the unannealed Au film/GaN, due to intimate interface contact, which is a key factor that enables efficient charge transfer and injection at the interfaces.”

7. The authors mention that for some measurements the gold film was annealed, forming gold nanoislands. This would result in a system more like the AuNP/GaN sample and should exhibit similar results. The authors show describe these comparisons and explain the results.

Response: Thanks for the reviewer’s comment. We would like to clarify that all results reported for the Au NP/GaN samples in our work correspond to gold films that were annealed to form gold nanoislands. In other words, the Au NP/GaN samples discussed throughout the manuscript are indeed the annealed gold films. The sample preparation process, including the annealing procedure, is described in detail in the Methods section in the revised manuscript.

8. The authors exclude contributions from Auger recombination based on a lack of a pump power dependence. However, the effect of Auger scattering is readily seen in TR-2PPEM literature showing a suppression of the d-band effect in the hot electron lifetime (doi:10.1103/PhysRevB.57.12812 and 10.1016/S0039-6028(96)01353-2). So, it is odd to

exclude such a mechanism from hot carrier generation and injection, especially at excitation wavelengths below the d-band onset. Also, Auger recombination results in the same athermal energy distribution as intraband excitation and given the excitation energy used interband excitation should be the dominate absorption mechanism.

Response: Thanks for the reviewer's insightful comment. We fully agree that Auger scattering can affect hot-carrier relaxation dynamics in noble metals, particularly in the presence of interband transitions, as discussed in the cited works. However, we note that the characteristic Auger-related spectral features are most prominently observed in Cu systems, including the references suggested by the reviewer and another paper (*Phys. Rev. B*, **1997**, *56*, 1099-1102), in which d-band holes predominantly decay via Auger-like processes. In contrast, Au possesses a deeper-lying d-band and exhibits weaker Auger decay channels, owing to the more effective screening provided by its spatially extended 5d orbitals compared with the more localized 3d orbitals in Cu (*Prog. Surf. Sci.* **2015**, *90*, 319-376). As a result, the d-band effect in carrier lifetimes is generally much less pronounced in Au than in Cu.

Consistent with these considerations, our TR-2PPE measurements on Au/GaN did not reveal the characteristic pump-fluence dependence (Supplementary Figure 12) or the spectral signatures in the lifetime-energy curve (Figure 3c) typically associated with Auger recombination. This observation is consistent with previous reports on Au system (*Phys. Rev. Lett.*, **2020**, *125*, 076803; *Phys. Rev. B*, **1998**, *58*, 10948-10952; *Prog. Surf. Sci.* **2015**, *90*, 319-376), where relaxation of d-band holes through Auger processes does not strongly impact the excited-electron dynamics. Based on this evidence, we conclude the Auger contribution to hot-carrier relaxation to be minor in our system.

9. It is surprising that the hot electrons injected into the GaN are so short lived. IR-transient absorption spectra of hot holes injected into p-type GaN exhibit injected carrier lifetimes of tens of picoseconds (10.1038/s41563-020-0737-1). While the authors explain the short lifetime is due to optical phonons, shouldn't one expect similar reasoning for injected hot holes?

Response: Thanks for the reviewer's insightful comment. We agree that injected hot carriers in GaN exhibit distinct relaxation times depending on their nature (electrons vs. holes).

First, considering the carrier type, the IR-transient absorption study cited by the reviewer (*Nat. Mater.* **2020**, *19*, 1312–1318) reported that injected hot holes in *p*-type GaN exhibit lifetimes on the order of tens of picoseconds, significantly longer than the ultrafast lifetimes (~100 fs) observed for injected electrons in our study. This asymmetry in carrier dynamics can be attributed to stronger coupling of conduction-band electrons to polar LO phonons, differences in phonon modes between the valence and conduction bands, and the smaller effective mass of electrons compared to holes (*Phys. Rev. B*, **2015**, *92*, 235439; *Appl. Phys. Lett.* **1995**, *67*, 1757–1759), all of which lead to faster energy dissipation and ultrafast relaxation for electrons relative to holes.

Second, the difference in observed lifetimes can also be attributed to the measurement techniques. The IR-transient absorption spectroscopy employed in the cited work primarily probes bulk carrier dynamics, capturing responses averaged over the entire material volume. In contrast, our TR-2PPE measurements are highly surface-sensitive, probing carriers within approximately the first 10 nm beneath the sample surface. Surface and interface regions often exhibit faster relaxation due to stronger electron–phonon coupling and possible contributions from surface states. Consequently, the shorter hot-electron lifetimes observed in our TR-2PPE study may partly reflect the enhanced relaxation dynamics near the surface and interface, in addition to the intrinsic differences between electron and hole relaxation pathways.

Taken together, both the intrinsic differences between electron and hole relaxation pathways and the measurement method differences explain the shorter hot-electron lifetimes observed in our surface-sensitive TR-2PPE study relative to bulk IR-transient absorption results.

Major Comments

1. While the gold film/GaN heterostructure seems to lack a strong injection dependence, the authors should have a control experiment of the gold nanoparticles/film on a material that lacks interfacial charge transfer such as SiO₂ or Al₂O₃ to compare the dynamics against. It is surprising that such a control was not included.

Response: We thank the reviewer for this valuable suggestion. We prepared control heterostructures by inserting an ultrathin Al₂O₃ interlayer (~5 nm) at the Au/GaN interface using atomic layer deposition. XPS characterization confirmed the successful formation of the insulating Al₂O₃ interlayer (Figure R5). TR-2PPE results on these control samples are shown in Figure R6,R7. Compared to the Au NP/GaN heterostructure, insertion of the Al₂O₃ layer did not significantly alter the initial hot-electron energy distribution showing nonthermal electron distribution extending up to $E_F+h\nu$, but it led to faster carrier energy relaxation. This behavior can be attributed to the suppression of interfacial charge transfer, which results in a higher density of hot carriers confined within the Au. The increased hot-electron population enhances electron-electron scattering events and accelerates electron relaxation (*ACS Nano* **2024**, *18*, 19077-19085). A similarly longer lifetime of plasmonic electrons has been observed in Au/TiO₂ compared to Au/SiO₂ systems based on the studies of TAS (*Nature Communications* **2024**, *15*, 703). For the Au film/GaN interface, insertion of the Al₂O₃ layer effectively suppressed the interface-state feature observed near ~1.5 eV, while leaving the hot-electron energy distribution and relaxation dynamics largely unaffected.

We have added these results in the revised SI (Supplementary Figs. S20,21) and incorporated the relevant discussion into the revised manuscript to strengthen the interpretation, as shown below:

(P14-P15) “As a reference for the hot electron dynamics, we also prepared a control sample by inserting an insulating Al_2O_3 interlayer at the Au/GaN interface (see Methods for details). TR-2PPE measurements show that the presence of Al_2O_3 does not alter the initial energy distribution of hot electrons but accelerates their energy relaxation (Supplementary Figs. S20,21). This effect arises from the suppression of interfacial charge transfer, which increases the hot-electron population within Au and thereby enhances electron–electron scattering⁵⁵, leading to faster cooling and shorter lifetimes.”

55. Lee, A., Wu, S.X., Yim, J.E., Zhao, B.Q. & Sheldon, M.T. Hot Electrons in a Steady State: Interband vs Intraband Excitation of Plasmonic Gold. *ACS Nano* **18**, 19077-19085 (2024).

Figure R5. Comparison of high-resolution XPS of (a) Ga $2p$ and (b) Al $2p$ for Au/GaN and Au/ Al_2O_3 /GaN heterostructures. Due to the low electron IMFP of around 2 nm for the 1486.74 eV excitation used, no clear Ga signal is observed from the Au/GaN surface after inserting a 5 nm Al_2O_3 interlayer. The Al $2p$ peak verifies the presence of the Al_2O_3 layer.

Figure R6. (a) Representative pseudo-color plot of TR-2PPE spectra of Au NP/Al₂O₃/GaN excited at 2.25 eV and probed at 4.33 eV. (b) Representative pseudo-color plot of Au film/Al₂O₃/GaN excited at 2.25 eV and probed at 4.33 eV. (c,d) Background-subtracted 2PPE spectra of Au NP/Al₂O₃/GaN (c) and Au film/Al₂O₃/GaN (d) samples at the positive delay times. (e,f) Background-subtracted 2PPE spectra of Au NP/Al₂O₃/GaN (e) and Au film/Al₂O₃/GaN (f) samples at the negative delay times.

Figure R7. (a) Energy-resolved photoelectron intensity as a function of delay time for Au NP/Al₂O₃/GaN samples excited at 2.25 eV. (b) Energy-resolved photoelectron intensity as a function of delay time for Au film/Al₂O₃/GaN excited at 2.25 eV. (c) Lifetimes of excited carriers as a function of electron energy relative to E-E_F for both Au NP/Al₂O₃/GaN and Au film/Al₂O₃/GaN samples.

Reviewer #3 (Remarks to the Author):

Response: We sincerely thank the reviewer for his/her time and consideration in evaluating our manuscript.

Responses to the reviewer(s)' comments

Reviewer #1 (Remarks to the Author):

The authors have clearly claimed all my comments and I would like to recommend its publication on Nature Communications as its current version.

Response: We sincerely thank the reviewer for the careful evaluation and are grateful for the positive feedback.

Reviewer #2 (Remarks to the Author):

The Authors have sufficiently addressed the reviewers' comments.

Response: We appreciate the reviewer's recognition of our responses and are glad that all comments have been addressed.